# Exploring App-Based Taxi Movement Patterns from Large-Scale Geolocation Data

**Wenbo Zhang** * and **Chang Xu**

School of Transportation, Southeast University, Nanjing 211189, China; 213190539@seu.edu.cn
* Correspondence: wenbozhang@seu.edu.cn; Tel.: +86-025-8379-2500

**Abstract:** This study is designed to leverage ubiquitous mobile computing techniques on exploring app-based taxi movement patterns in large cities. To study patterns at different scales, we comprehensively explore both occupied and unoccupied vehicle movement characteristics through not only individual trips but also their aggregations. Moran's $I$ and its variations are applied to explore spatial autocorrelations among different rides. PageRank centrality is applied for a functional network representing traffic flows to discover places of interest. Gyration radius measures the scope of passenger mobility and driver passenger searching. Moreover, cumulative distribution and data visualization techniques are adopted for trip level characteristics and features analysis. The results indicate that the app-based taxi services are serving more neighborhoods other than downtown areas by taking large proportion of relatively shorter trips and contributing to net increase in total taxi ridership although net decrease in downtown areas. The spatial autocorrelations are significant not only within each service but also among services. With the smartphone-based applications, app-based taxi services are able to search passengers in a larger area and move more efficiently during both occupied and unoccupied periods. Mining from huge empty trip trajectory by app-based taxis, we also identify the existence of stationary/stops state and circulations.

**Keywords:** traditional street-hailing taxicabs; emerging app-based taxi services; spatiotemporal movement patterns; occupied and unoccupied vehicle movements

## 1. Introduction

The smartphone application-based taxi services (also called e-hailing or transportation network companies), such as Uber and Lyft, have grown exponentially in recent years and are challenging the predominance of traditional street-hailing taxicabs. Although both app-based and traditional taxi services provide similar door-to-door urban mobility, app-based taxi services have a few extraordinary features that make them competitive: First, the smartphone applications adopted by app-based taxi services connect passengers with driver partners and provide real time feedback that can help learn patterns surrounding demand or supply; second, the free supply model allows driver partners to begin or end services anywhere and anytime, thus attracting considerable part-time driver partners; third, pricing can be adjusted dynamically by a multiplier that is developed based on local demand and supply; and fourth, various services are included in one single application, which can meet requirements by different groups, for instance, family trips with children, disabled, luxury trips, and economic trips. Taking these advantages, on one hand, passengers are more informative on their taxi trips, including how many minutes they should wait, how much the trips cost, and which service are available, instead of trying your luck on street-side especially in bad weather. On the other hand, driver partners can choose appropriate times and locations for service to maximize their revenues. All these innovations are reshaping the urban taxi market. For example, app-based taxi ridership doubled annually over the last three years to 133 million passengers in 2016 and is approaching traditional taxicab ridership level in New York City (NYC), where is the largest taxi market

in North America operating more than 13,000 yellow taxicabs and experiencing severe declines in ridership, almost reduced by half in the last few years [1].

Beyond the changes in ridership, another key issue is whether the app-based taxi service disturbs current mobility patterns by traditional taxicabs. Understanding of mobility patterns plays an important role in further in-depth discussions on taxi market modeling and regulations. A comprehensive empirical study of the following questions is necessary: How do the informative app-based taxi driver partners search for passengers, compared with traditional uninformative taxicab drivers? How will the introduction of app-based taxi services redistribute demand or flows across the whole city? How much more efficient are the app-based taxicabs? What special travel characteristics does an app-based taxi service have that can differentiate them from traditional taxi services? Moreover, it has been approved that the app-based taxi service has generated net increases of 31 million trips and 52 million passengers in NYC since 2013, after accounting for declines in traditional services [1]. Do these new passengers introduce new mobility patterns that make app-based taxi services different? Although endless debates over the emerging controversial taxi services have attracted much attention from various domains, such as economists, engineers, policy makers, and urban planners, little hard evidence exists to support either position, i.e., whether there has been change or no change.

Analyzing mobility patterns is a critical but difficult topic, since it is important for addressing many urban sustainability challenges, and always covers complicated spatiotemporal patterns generated from interconnected heterogeneous groups of individuals. However, ubiquitous mobile computing and the massive data it generates present new opportunities to advance our understandings of the complex dynamics in urban system. For example, mobility data have become increasingly available due to the extensive usage of location-aware techniques, which consist of a set of moving objects and their trajectory or time-stamped locations. The datasets such as huge taxi trajectory are always with better spatial and temporal coverage, compared with traditional survey-based transportation dataset, and are extensively integrated into studies on taxi movement patterns [2–5].

To understand app-based taxi movement patterns for both occupied trips with passengers and unoccupied trips for customer-searching, this study will fully leverage current advances in ubiquitous mobile computing techniques along with real datasets of Uber (i.e., the most successful app-based taxi service) and yellow taxicab (i.e., a traditional street-hailing taxi service) mobility dataset in NYC. The patterns are discussed in four major ways: (1) spatial distribution of aggregated trip flows and places of interest; (2) trip displacement patterns in terms of travel time and distance; (3) vehicle movement efficiency across areas; and (4) characteristics of Uber driver partners' passenger searching movements. Our main contributions are two-fold: First, this is one of the first few empirical studies on app-based taxi movement patterns and their comparisons with traditional taxis; and second, this study conducts a multi-scale and multi-facet analysis on movement patterns, from flows at the aggregate level to trajectory at the trip level and from occupied trips to unoccupied trips. The remaining sections are organized as follows: the second section summarizes current literature on taxi movement patterns; the third section presents real datasets and preprocessing; the fourth section explores occupied trip movement patterns; the fifth section discusses unoccupied trip movement and special analyses on Uber passenger searching behaviors; and the last section concludes empirical findings.

## 2. Literature Review

Recent work has made great progress in understanding travel patterns of traditional taxicabs with increasing availability of taxi mobility data. Most work is developed along four main ways: First, analyzing general temporal/spatial or spatiotemporal patterns of taxi movements with common statistical properties. Cai et al. combined occupied and unoccupied taxi trips together as one integrated system, instead of focusing on only occupied or unoccupied trips, and identified spatial and temporal regularities of travel time and travel distance in taxi travels [3]. Wang et al. explored the occupied taxi trips on

holidays through comparing spatial distribution of pickups and drop-offs, trip displacement, and spatial scope of taxi activities with trips on regular days [6]. On the contrary, Peng et al. focused on millions of weekday taxi trips, extracting three main features by trip purpose that can approximate all taxi trips in a linear combination, and identifying spatiotemporal variations of each feature at the small geographical unit level [7]. Zong et al. uncovered unoccupied taxi trips over multiple days through measuring inter-daily variations and checked differences in the drivers' learning ability and routine behaviors [8]. Second, it is possible to figure out subgroups of interests and analyze travel patterns of those subgroups. Kumar et al. [5] and Matsubara et al. [9] applied clustering techniques on taxi trip OD points and trajectory to extract featured movements, respectively. Guo et al. clustered regions by taxi ride origins and destinations and applied spatiotemporal pattern analysis on in- and out-flows in each cluster [10]. Noboa et al. [11] and Qian et al. [2] related the spatiotemporal clusters of taxi rides with land use or points of interest and identified featured taxi movements with trip purpose. Moreover, Ding et al. explored long break status along each empty taxi trip through clustering corresponding trajectory points and identified spatiotemporal patterns of those long breaks during taxi customer searching [12]. In addition to spatial clustering methods, Dong et al. [13] and Tang et al. [14] extracted top drivers by levels of income or customer searching efficiency and discovered spatiotemporal patterns of both top drivers and other drivers. Third, representing taxi movements as a network. Hoque et al. [15] and Pascual [16] transformed taxi trips to a weighted directed complex network and introduced various network metrics, for instance, node correlations, assortativity, clustering, and degree of connectivity, to reveal topological information in the taxi system. Last, interactions with other transportation modes can be studied. Wang and Ross divided taxi rides into three types based on spatial similarity with sections of the subway network and examined travel patterns of each type of taxi rides, as well as associations with built environment [17]. Li et al. utilized both taxi trajectory data and subway transactions to empirically measure impacts of new subway lines on the spatiotemporal distribution of taxi rides [18].

Compared to traditional taxicab movements, empirical studies on app-based taxi trips are much fewer and adopted methodologies have less variety, which is limited by data availability. Existing analyses are obtaining app-based taxi trips in two ways: First, they collaborate with transportation network companies and then implement basic statistical analyses. Hall and Krueger combined survey data with administrative data provided by Uber and explored socioeconomic status, work durations, and earnings of Uber driver partners across US major cities [19]. Cramer and Krueger measured the movement efficiency of UberX and traditional taxicabs in terms of fraction of times and share of miles with a passenger, respectively, and concluded that UberX are more efficient based on cumulative distribution of capacity utilization rates. They also listed four possible reasons for higher UberX utilization rate: more efficient driver passenger matching technology, Uber's city scale platform enabling faster matches, inefficient taxi regulations, and Uber's flexible labor supply and surge pricing, but failed to provide hard evidence [20]. Second, researchers can collect data directly from client applications then apply statistical analyses. Schwieterman and Michel completed 50 trips by UberPool and Chicago transit, respectively, and concluded that UberPool was an option attractive to far more than extremely time conscious travelers, not to many commuters, and UberPool tended to perform worst while traveling from/to central business district due to heavy traffic [21]. This study provided few empirical efficiency comparisons between UberPool and Transit, but was highly limited by data coverage. Current progress in web crawlers presents a new opportunity for obtaining data directly from client applications, which can greatly improve data coverage in an efficient way. Chen et al. developed an emulation tool that can obtain information displayed on an Uber passenger's client application, including up to eight available vehicles' trajectory information and current pricing structure. Although the tool only covered the core of Manhattan, the empirical findings from the dataset provided insights on impacts of dynamic pricing on Uber demand and supply [22]. Guo et al. also wrote a script that can

track Shenzhou (a Chinese app-based taxi platform) users' behavior while requesting a ride, which can enable them to explore the passengers' reaction to dynamic pricing while requesting and measure occupied trip displacement, as well as spatial distribution and surge multiplier generation [4,23].

To sum up, there are extensive discussions on traditional taxicab movements, covering occupied taxi trips, empty taxi trips, frequent flows, top drivers, and spatiotemporal patterns. However, we still lack perfect and comprehensive understanding of emerging app-based taxi movement, especially their differences with traditional taxicab movement. The information techniques greatly improve data availability of app-based taxis, which can yield huge mobility data as compared to traditional taxicabs. This further enables us to leverage current progresses in ubiquitous mobile computing techniques on empirically uncovering movement patterns based on high-resolution mobility data.

## 3. Data Description

### 3.1. Data Source

We selected yellow taxicab, Boro taxicab, and UberX in NYC (all three labeled regions in Figure 1) as observations due to following reasons: (1) NYC has the largest taxi market in North America, as well as one of largest Uber markets in US. The Uber ridership is approaching to the traditional taxicab rides. (2) Yellow and Boro taxicabs are two main traditional street-hailing taxi services. The only difference is the legal pickup area that Boro taxicab can only pickup outside downtown and midtown Manhattan but yellow taxicabs do not have the limitations. (3) Uber has a much higher app-based taxi market share and is the dominant company. UberX is the economic and most popular product by Uber. The yellow and Boro taxicab trip records are open access to the public through the official website of NYC Taxi Limousine and Commission (NYCTLC). However, the official dataset does not record any identity information such as taxicab medallion id (i.e., a unique permit assigned to each cab) or driver id (i.e., issued taxicab driver license id), due to privacy protection. Thus, we refer to an additional taxicab dataset from Illinois Data Bank [24] that is a FOIL dataset from NYCTLC but with anonymous vehicle and driver identify information. The additional dataset can help us estimate taxicab empty trips for each driver and compare empty taxicab movement patterns with Uber empty vehicles.

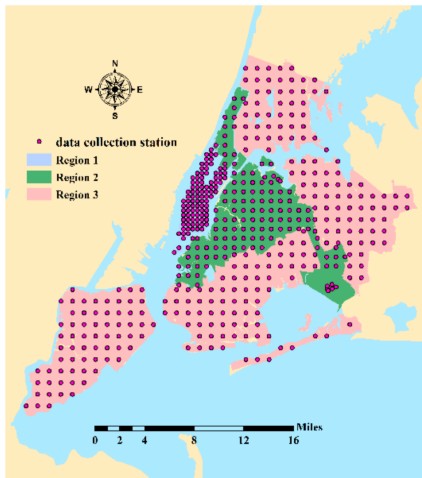

**Figure 1.** The selected regions and corresponding data collection stations.

Except for obtaining traditional taxicab mobility dataset, the main objective of data preparation is to develop a web crawler that can obtain reliable Uber mobility dataset with good spatial and temporal coverage. Inspired by Chen et al. [22], we develop a similar tool for Uber data collection. The difference is, however, that we are able work and extract directly from the Uber Mobile platform (a web version of Uber client app, ref: https://m.uber.com, accessed on 12 July 2021) instead of the smartphone-based application.

More importantly, we can extend our data collection to the whole of NYC, not just the core of Manhattan, through deploying many more data collection stations across NYC; about 470, as shown in Figure 1. Finally, we run the tool every 5 a.m. to midnight starting from 8 April to 1 May 2017. The obtained raw records are presented in Table 1.

### 3.2. Data Preprocessing

Before processing all three raw datasets, we should clarify two temporal scale problems on the three data samples. First, UberX dataset is collected from 8 April to 1 May 2017, yet NYCTLC has not been released for taxicab trip records in this April. Thus, we take yellow and boro taxicab trip records from 2 April to 25 April 2016 to represent current taxicab service condition. Second, also limited by data availability, the latest datasets of yellow taxicab trip records with vehicle and driver ids are in 2013. Thereafter, NYCTLC does not provide any datasets with driver or vehicle information with the public. Thus, we take yellow taxicab trip records from 6 April to 29 April 2013 to explore empty taxicab travel patterns. Boro taxicab was not issued in the month thus it is not included. Although Yellow taxicab decreased a lot in ridership from 2013 to 2016, the utilization rate remains steady [25] and the 2013 dataset reflects empty taxicab movement patterns in 2017 to a certain extent. Moreover, the introduction of yellow taxicab trips in 2013 can allow us to measure temporal variations in taxicab travel patterns.

The first preprocessing is an active Uber driver filter. We observe about 5 million unique driver ids but over 4.5 million driver ids only serve in fewer than 5 days, mostly only in one single day. Thus, we think most driver ids are inactive and only keep active drivers of interest who serve for more than 10 days. Applying this filter, we obtained a set of about 43,000 active driver ids that is almost same as the fleet of dispatched Uber drivers in the for-hire vehicle base aggregate weekly report by NYCTLC (ref: http://www.nyc. gov/html/tlc/html/technology/aggregated_data.shtml, accessed on 12 July 2021). The matched numbers provide strong evidence that we very likely capture the whole Uber fleet.

The second preprocessing is the empty or occupied trips estimation. The Yellow taxicab trip dataset in 2013 only contains occupied trips. We should refer to driver id to estimate empty trips. Sorting by unique driver id and occupied trip start time, as well as checking differences in two sequential start times of occupied trips, enable us to generate a sequence of occupied trips in one same shift. Considering taxicab regulations that one shift should serve for at least 8 h without intermediate stops or breaks, we think that one empty taxicab trip may exist between two sequential occupied trip records and the empty trip information can be estimated based on destination of one occupied trip and origin of next occupied trip by one same driver, shown in Table 1. In addition, Uber empty and occupied trips should be estimated from Uber vehicle trajectory through comparing time and distance differences between two sequential trajectory points by one same vehicle: (1) empty trips: if the time difference is less than 60 s, or if the time difference is greater than 60 s but less than 2 h and the distance difference is less than 400 m; (2) occupied trips: if the time difference is greater than 60 s but less than 2 h and the distance difference is greater than 400 m; (3) offline or stop service: if the time difference is greater than 2 h.

The last preprocessing step is trip aggregation. Instead of working directly with massive points in terms of longitude and latitude, the rides are aggregated at census tracts that are small subdivisions of a city and provide a stable set of geographic units for presentation of statistical data. This process can enable us to analyze travel patterns at not only the trip level but also the aggregate level. In NYC, there are about 2164 census tracts, more than 70% of which have the area less than 0.1 square miles (similar to a 500 m by 500 m grid).

**Table 1.** Summary of collected datasets and preprocessing.

| Services | Duration | Records | Preprocessing |
|---|---|---|---|
| Yellow and Boro Taxicab | 2 April to 25 April 2016 | Occupied trips: $\left\{ t^i_O, t^i_D, \mathrm{lat}^i_O, \mathrm{lat}^i_D, \mathrm{lng}^i_O, \mathrm{lng}^i_D, d^i, f^i \right\}_{i \in I}$ | Occupied trip travel time: $tt^i = t^i_D - t^i_O$ <br> Occupied trip aggregation: $CT^i_O, CT^i_D$ |
| Yellow Taxicab | 6 April to 29 April 2013 | Occupied trips: $\left\{ vid_j, did_j, t^j_O, t^j_D, \mathrm{lat}^j_O, \mathrm{lat}^j_D, \mathrm{lng}^j_O, \mathrm{lng}^j_D, d^j, f^j, tt^j \right\}_{j \in J}$ | Occupied trip aggregation: $CT^j_O, CT^j_D$ <br> Empty trips extraction: <br> $\left\{ vid_j, did_j, t^j_D, t^{j+1}_O, \mathrm{lat}^j_D, \mathrm{lat}^{j+1}_O, \mathrm{lng}^j_D, \mathrm{lng}^{j+1}_O, tt^j = t^{j+1}_O - t^j_D, CT^j_D, CT^{j+1}_O \right\}_{j \in E(J)}$ |
| UberX | 8 April to 1 May 2017 | Trajectory of empty vehicles: <br> $\left\{ vid_k, pid_k, t_k, dir_k, \mathrm{lat}_k, \mathrm{lng}_k \right\}_{k \in K}$; <br> Surge pricing: <br> $\left\{ loc_s, pid_s, t_s, mult_s, mEta_s, aEta_s, et_s \right\}_{s \in S}$ | Uber empty trips: <br> $\left\{ vid_m, pid_m, t^m_O, t^m_D, \mathrm{lat}^m_O, \mathrm{lat}^m_D, \mathrm{lng}^m_O, \mathrm{lng}^m_D, tt^m, CT^m_O, CT^m_D \right\}_{m \in M}$ <br> Uber occupied trips: <br> $\left\{ vid_q, pid_q, t^q_O, t^q_D, \mathrm{lat}^q_O, \mathrm{lat}^q_D, \mathrm{lng}^q_O, \mathrm{lng}^q_D, tt^q, CT^q_O, CT^q_D \right\}_{q \in Q}$ |

Note: $t_O$, trip start time; $t_D$, trip end time; $\mathrm{lat}_O$, latitude of trip start location; $\mathrm{lat}_D$, latitude of trip end location; $\mathrm{lng}_O$, longitude of trip start location; $\mathrm{lng}_D$, longitude of trip end location; $CT_O$, census tract of trip start location; $CT_D$, census tract of trip end location; $d$, metered distance in miles; $f$, charged fare in dollars; $tt$, trip travel time in seconds; $vid$, vehicle id; $did$, driver id; $pid$, Uber product id; $t$, current time; $dir$, vehicle direction in a degree of 360; $lat$, latitude of trajectory point; $lng$, longitude of trajectory point; $loc$, data collection station; $mult$, surge price multiplier; $mEta$, minimum estimated passenger waiting time in minutes; $aEta$, average estimated passenger waiting time in minutes; $et$, expiration time of current surge pricing in seconds; $I$, set of all records in file of yellow and boro taxicab; $J$, set of all records in the file of yellow taxicab; $E(J)$, set of occupied trips followed by empty trips; $K$, set of all Uber empty vehicle trajectory points; $S$, set of all surge pricing observations; $M$, set of all Uber empty trips; and $Q$, set of all estimated Uber occupied trips.

## 4. Occupied Movements

### 4.1. Aggregate Flow Patterns

4.1.1. Distribution of Generations and Attractions

From the three datasets, we observed 8,350,007 (including 952,611 Boro Taxicab Occupied Trips) Taxicab occupied trips in 2016, as well as 9,242,133 Taxicab occupied trips in 2013, versus 2,017,500 Uber occupied trips. At the city scale, the app-based taxi not only competes with taxicab for passengers, considering a significant decline in ridership in the last few years, but also induces new passengers, considering a net increase in total ridership of all taxi services. Figure 2 presents the spatial distribution of pickups and drop-offs, as well as temporal variations, between 2013 and two recent years. Both traditional and app-based taxis show similar spatial distribution that mainly concentrates at downtown areas and two airports. The minor difference is that traditional taxicabs pick up more passengers along the highway among two airports and Manhattan Island, but app-based taxis take a few more passengers from/to remote areas, such as Staten Island and the upper part of the city, close to the city boundary. The net increases in traditional and app-based taxis indicate that most new demand is generated from or travels to outside Manhattan, especially downtown areas of Brooklyn and Queens. Moreover, app-based taxis are more popular than traditional taxicabs for these new demands, based on the evidence of a higher net increase in pickups and drops if we combine both taxis but a much smaller net increase if we only compare traditional taxicabs. More interestingly, both trips to/from downtown areas by both taxis, i.e., Manhattan, are experiencing significant decline, compared with ridership in 2013. This is likely due to severe traffic congestion, more delays, and popularity of shared bikes and other modes. Lastly, the trips also redistribute while traveling to/from two main airports, JFK (i.e., the remote one at lower right corner) and LGA (i.e., the one close to Manhattan). Traditional taxis are serving many more trips from/to JFK but experiencing higher decline at LGA. This does not mean there is a declining demand for taxi trips at LGA, since app-based taxis take those decreasing trips by traditional taxis but also induce new demands at LGA.

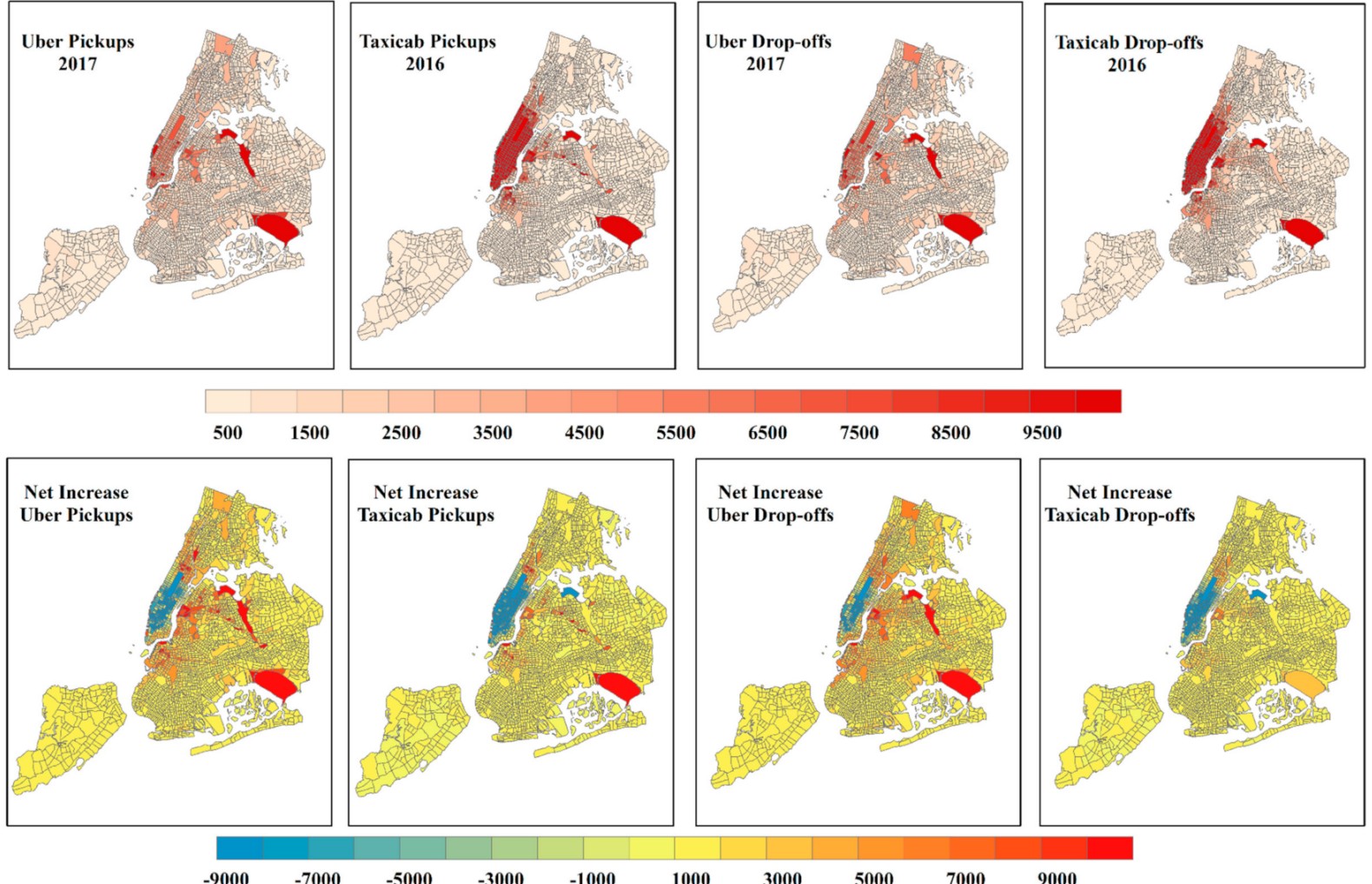

**Figure 2.** Spatial distributions of pickups and drop-offs (Net increase Uber pickups: difference between the summation of Uber 2017 and Taxicab 2016 and Taxicab 2013; Net increase Taxicab pickups: difference between Taxicabs in 2016 and 2013).

### 4.1.2. Local Spatial Autocorrelations

This section further checks the spatial relationship of taxi rides, including relationship of the rides in one unit with rides by the same service in surrounding units and relationship of the rides in one unit with rides by different taxi service in surrounding units. Here, we introduce a very common statistic global Moran's $I$ and its variations of local Moran's I, as well as Bivariate local Moran's $I$ [26–28]. The global Moran's $I$ is designed to measure spatial autocorrelations at the system level through combining all spatial effects of system units, shown in Equations (1)–(4). In this case, the negative value of Moran's $I$ means one unit with high/low taxi rides is always surrounded by units with low/high taxi rides and inversely the positive value of Moran's $I$ means one unit with high/low taxi rides is always surrounded by units with high/low taxi rides. Thus, positive spatial autocorrelation always shows clusters on maps, such as the taxi rides distribution. Equations (5) and (6) present the bivariate version of Moran's $I$, which allows us to analyze the relationship between traditional taxicab rides in surrounding neighborhood and app-based taxi rides, and vice versa.

$$I = \frac{n}{S_0} \frac{\sum_{i=1}^{n} \sum_{j=1}^{n} w_{i,j} z_i z_j}{\sum_{i=1}^{n} z_i^2} \tag{1}$$

$$S_0 = \sum_{i=1}^{n} \sum_{j=1}^{n} w_{i,j} \tag{2}$$

$$z_i = \frac{x_i - \overline{x}}{\sigma_x} \tag{3}$$

$$I_i = \frac{(n-1)z_i \sum_{j \in N(i)} w_{ij} z_j}{\sum_{j \in N(i)} z_j^2} \tag{4}$$

$$I^{xy} = \frac{n}{S_0} \frac{\sum_{i=1}^{n} \sum_{j=1}^{n} w_{i,j} z_i^x z_j^y}{\sum_{i=1}^{n} \left(z_i^x\right)^2} \tag{5}$$

$$I_i^{xy} = \frac{(n-1)z_i^x \sum_{j \in N(i)} w_{ij} z_j^y}{\sum_{j \in N(i)} \left(z_j^y\right)^2} \tag{6}$$

where $I$ and $I_i$ are the global and local Moran's $I$ value and superscript $xy$ is a bivariate form of variable $x$ and variable $y$; $w_{i,j}$ is the spatial weight between two spatial units of $i$ and $j$; n is the number of spatial units; $z$ is the deviation from the variable mean value and superscript denotes variable; and $N(i)$ is the neighborhood of interest of spatial unit $i$.

As expected, all global Moran's $I$ statistics are positive, with 0.402 for app-based taxi rides in 2017, 0.7968 for traditional taxi in 2016, and 0.444 for both, which indicates the existence of significant spatial autocorrelation. Figure 3 further shows units with significant spatial autocorrelations and how they interact with surrounding units. Both traditional and app-based taxis, as well as their combinations, have similar spatial autocorrelation that units in Manhattan and remote areas are always with similar rides as their neighbors. Figure 3b classifies relationships into four categories: high-high, low-low, high-low, and low-high. The first two categories indicate positive spatial autocorrelation and the last two categories present negative spatial autocorrelation. Manhattan areas are always with a high-high pattern, yet remote areas are always with a low-low pattern. Other than this, app-based taxis also have a few special characteristics, for instance, a high-high pattern at

Brooklyn and Queens downtown and high-low pattern at a few remote areas, not shown by traditional taxicabs.

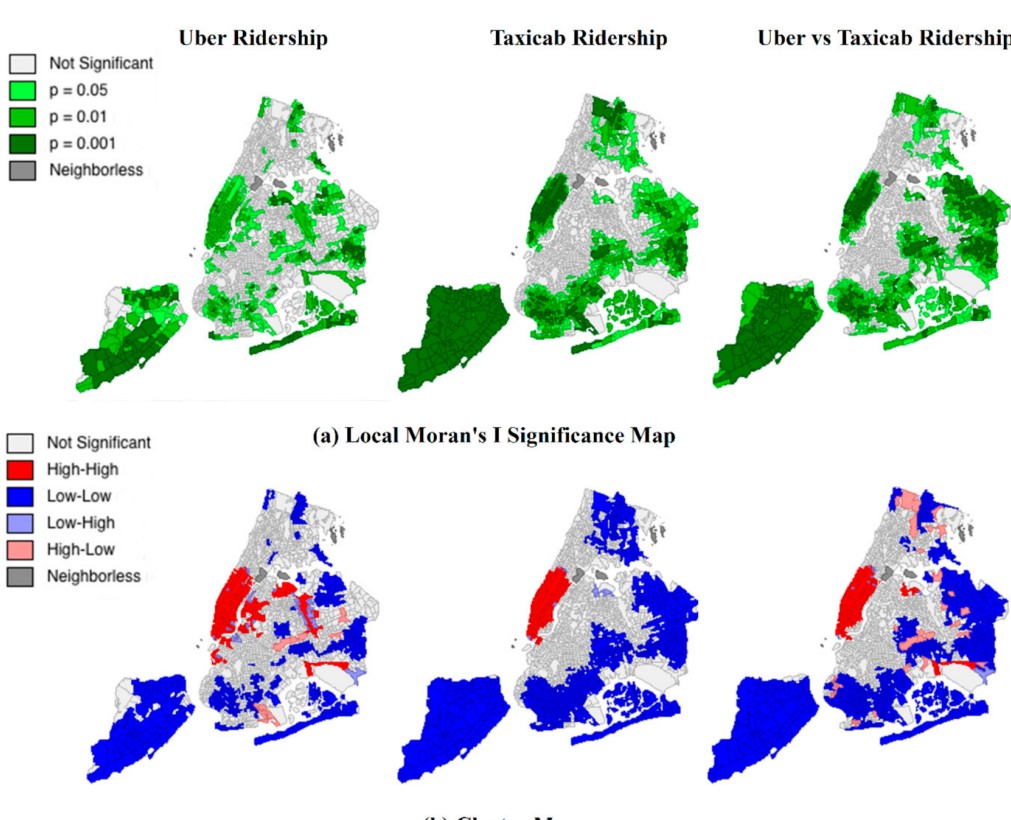

**Figure 3.** Local Spatial Autocorrelations of Uber, Taxicab, and Uber vs. Taxicab.

### 4.1.3. Places of Importance

Instead of studying on all spatial units, places of interest attract much more attention in transportation systems. This section introduces the idea of a complex network and represents taxi flows as a functional network with spatial units as nodes and taxi flows as weighted directed links, shown in Figure 4a. The degree of centrality based on weighted PageRank is adopted to measure the importance of each spatial unit. The key idea behind the definition of weighted PageRank centrality is that the spatial unit with the same number of outgoing or incoming flows (i.e., node degree in network) may not have the same importance in the flow network. Generally, an incoming flow from a strongly connected spatial unit is treated as more important than from a spatial unit with just a few connections with other spatial units and links with much more traffic flows are more important than those with just a few flows [15,16,29,30]. The mathematical forms are shown in Equations (7)–(9). Figure 4b states the ranks of each spatial unit by different taxi services. We can see that the ranks of spatial units in the app-based taxi flow network are significantly different from those in the taxicab flow network, by relative higher ranks of remote spatial units. App-based taxis provide an alternative and better transportation mode for those remote areas to connect with other hotspots. The ranks of spatial units in both taxicab flow networks remain almost stable. The only significant difference is that higher ranks of spatial units are more likely to cluster together in 2016. In brief, the ranks in all three flow networks do not vary greatly, which are Manhattan areas, followed by airports, Brooklyn and Queens downtown, and other remote areas.

$$W_{(v,u)}^{in} = \frac{N_u^I}{\sum\limits_{p \in R(v)} N_p^I} \tag{7}$$

$$W^{out}_{(v,u)} = \frac{N^O_u}{\sum\limits_{p \in R(v)} N^O_p} \tag{8}$$

$$PR(u) = (1-d) + d \sum_{v \in B(u)} PR(v) W^{in}_{(v,u)} W^{out}_{(v,u)} \tag{9}$$

where $W^{in}_{(v,u)}$ and $W^{out}_{(v,u)}$ are the weight of link between $v$ and $u$ calculated based on inflows and outflows respectively; $N$ is taxicab or Uber flows or number of trips, superscript denotes Inflows or Outflows, and subscript denotes census tract; $R(u)$ is the set of census tracts that census tract $u$ has flows to; $PR(*)$ is the PageRank value of one census tract; $d$ is a dampening factor that is usually set to 0.85; and $B(u)$ is the set of census tracts that have flows to census tract $u$.

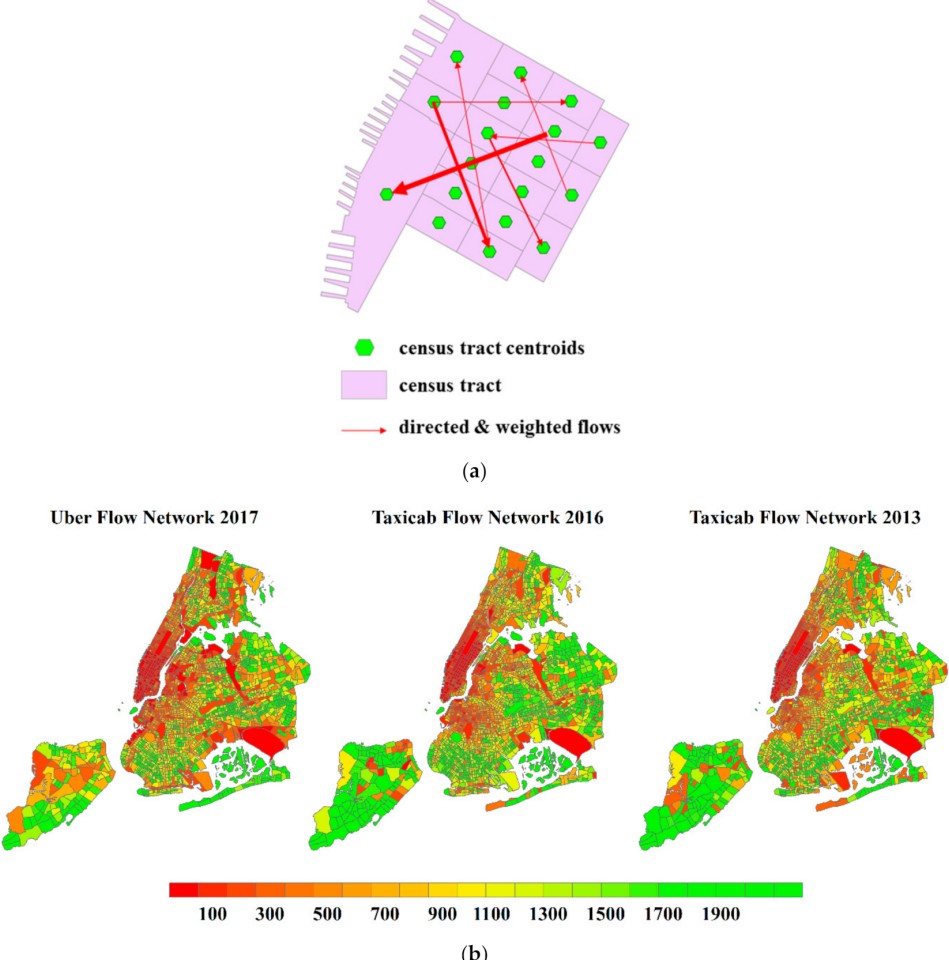

**Figure 4.** Importance of spatial units in aggregated trip flow networks (from red to green: high importance/rank to lower importance/rank). (**a**) Illustration of aggregated flow network representation, (**b**) Importance of locations ranked by PageRank.

### 4.1.4. Spatial Scope of Passenger Activities

Gyration radius is a metric to estimate the scope of activities [3,6,16]. Here, we refer to the idea for passenger movement scope by taxi from each unit, as shown in Equation (10). Figure 5 summarizes the hourly average gyration radius of each unit, as well as hourly standard deviation. From almost all units, passengers choose to take app-based taxis within a short radius of 2 to 6 miles with very small temporal variations. In Manhattan, traditional taxis also show a similar gyration radius distribution. However, they may drive passengers for a distant radius of more than 10 miles with small hourly variations. Moreover, the

passenger movement radius by traditional taxis remains stable from 2013 to 2016, except for few remote units where are with very few rides and radius are more likely to be influenced by occasional trips.

$$r_{g(i)} = \sqrt{\frac{1}{n_i} \sum_{j \in O(i)} \left( r_j - r_{c(i)} \right)^2} \qquad (10)$$

where $r_{g(i)}$ is the gyration radius of census tract $i$; $n_i$ is the number of trips from census tract $i$; $O(i)$ is the set of trips originated from census tract $i$; $r_j$ is the recorded trip $j$ end location; and $r_{c(i)}$ is the center of mass of all trip start locations in census tract $i$.

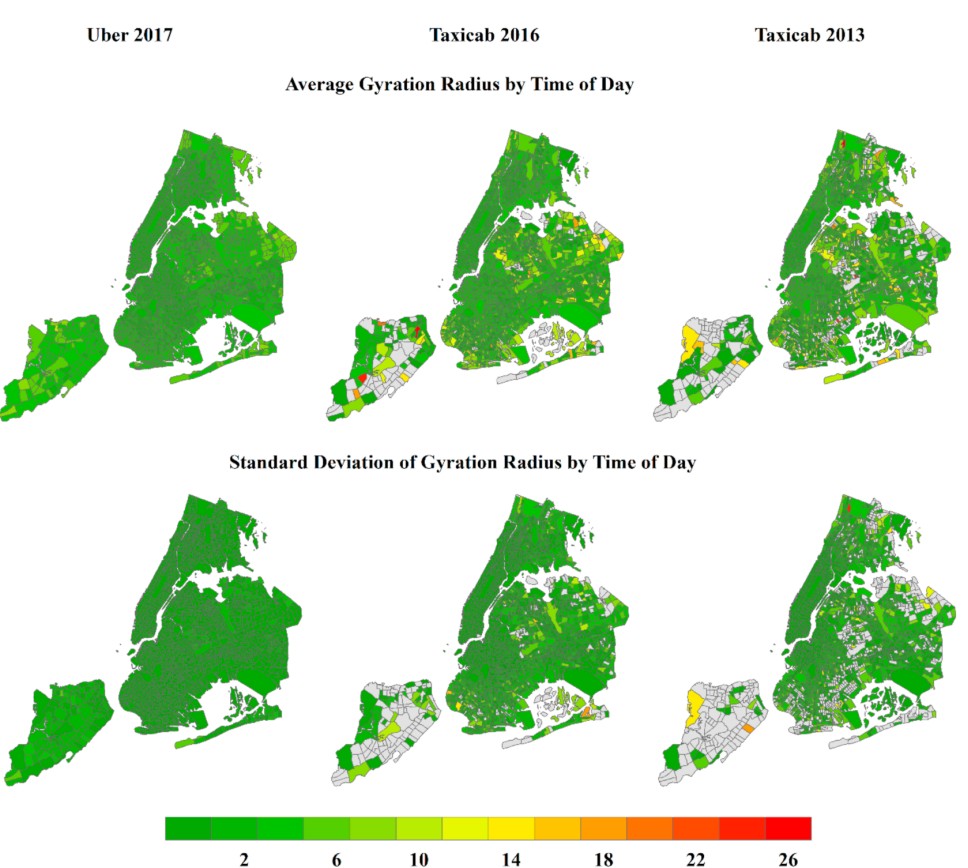

**Figure 5.** Gyration radius (miles) of spatial units.

### 4.2. Trip Travel Patterns

Other than the aggregated flow patterns, we also explore the general scaling properties of occupied trips and their movement efficiency. On one hand, the general scaling properties are obtained from cumulative distribution of travel time and trip distance. Here, the occupied app-based taxi travel distance is estimated based on the Euclidean distance between O and D, due to lack of reported occupied trips. The underlying assumption is that app-based taxis are designed to transport passengers to a destination as fast as possible thus likely choose the shortest route. On the other hand, we refer to Google Maps Direction API for benchmark travel time and distance then compute occupied trip movement efficiency. Due to the API's usage limits, we divide hours into rush hours (7–9 a.m. and 3–7 p.m.) and off rush hours, collect around 100 observations for each time slot, and estimate two travel time distributions for each of most frequent 50,000 census tract-based OD pairs. The sampling OD pairs cover about 78%, 74%, and 38% of occupied traditional taxicab trips in 2013 and 2016 and occupied app-based taxi trips in 2017, respectively.

Figure 6a presents the cumulative distribution of travel time and trip distance. For the traditional taxicabs, it remains stable from 2013 to 2016, although occupied trips in

2016 tend to have slightly longer travel times. The distribution of app-based taxi trip distance shows a slight difference, in that app-based taxis have more short trips less than 2–3 miles and fewer trips between 3 and 8 miles. Surprisingly, occupied app-based taxi trips seem to have a much higher travel time and yield a different distribution of travel time. The measurement errors may contribute to the condition of having a similar travel distance distribution but different travel time distribution. The Euclidean distance likely underestimates real displacement of occupied app-based taxi trips. Moreover, the measured travel time is the duration between ride request acceptance and ride served, including not only en-route travel time but also travel time to the ride request location after acceptance and waiting for passengers after early arrival. The last two times are almost negligible for traditional taxicabs. This may also increase the occupied app-based taxi trip travel time.

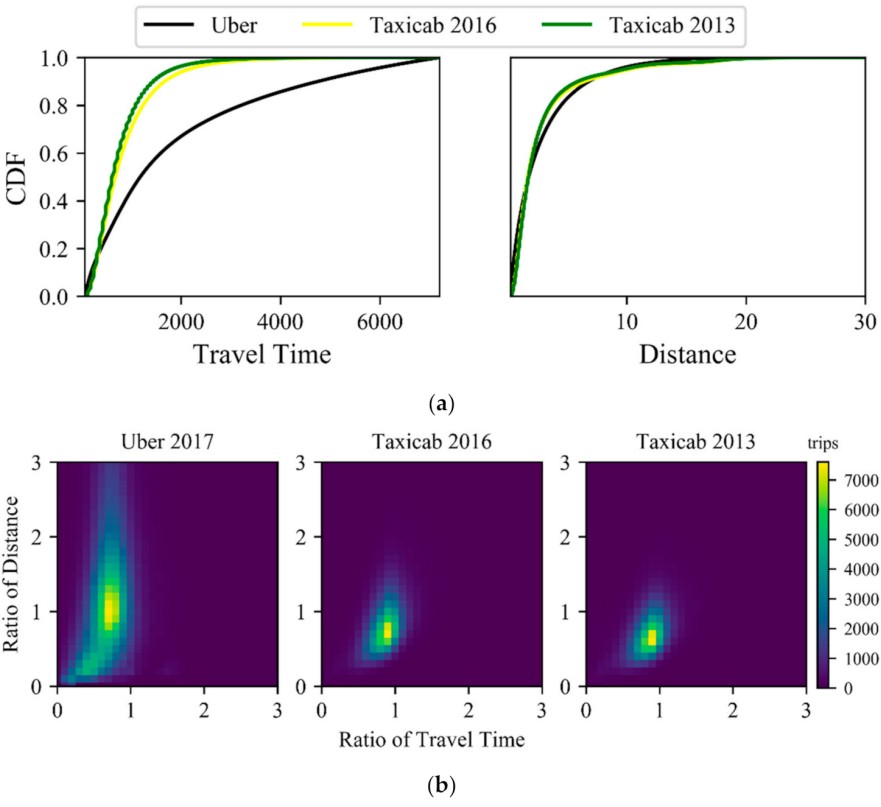

**Figure 6.** Cumulative distribution of trip displacement and movement efficiency. (**a**) Cumulative distribution of travel time (s) and distance (miles), (**b**) Occupied trip movement efficiency.

Figure 6b explores the movement efficiency through relating travel distance ratio to travel time ratio. Here, the travel time ratio is the ratio of measured travel time to expected Google travel time and travel distance ratio is the ratio of measured travel distance to expected Google travel distance (note that Google travel distance may differ depending on fastest travel time). The graph can reveal not only each trip efficiency in terms of distance and time separately, but also the speed comparison with common auto traffic. The diagonal line represents the same speed as Google common auto traffic and above or below the line means faster or slower than Google common auto traffic, respectively. Most traditional taxicabs perform identically (see the cluster in the right two figures) and can send their passengers efficiently almost as Google common auto traffic. From 2013 to 2016, traditional taxicabs also show a slight upward shift in movement efficiency. App-based taxi driver partners are, however, comparatively more efficient than both traditional taxicabs and Google common auto traffic. Although most app-based taxi trips have almost the same trip distance as Google (ratio of distance is 1), they can take less travel time to transport passengers to there, about 70–80% of Google's expected travel time. Surprisingly, few

app-based taxi driver partners may detour (see ratio of distance between 1 and 1.5) but they can still take less travel time to their destination, also being about 70–80% of Google's expected travel time.

## 5. Unoccupied Movements

### 5.1. Aggregate Flow Patterns

#### 5.1.1. Places of Importance

Same as analyses for occupied trips, we also represent empty traditional and app-based taxi trips as functional flow networks, respectively and apply PageRank centrality method. The weights of network links are number of empty trips. Similarly, Manhattan is still a hotspot for customer finding in Figure 7. However, app-based taxi driver partners likely extend their customer searching to a larger region, for instance, Brooklyn downtown and Bronx. Different from high importance in occupied trip network, airports are not first options for both taxis to search next passengers. This is likely due to potential longer waiting times at airports and remote locations of airports.

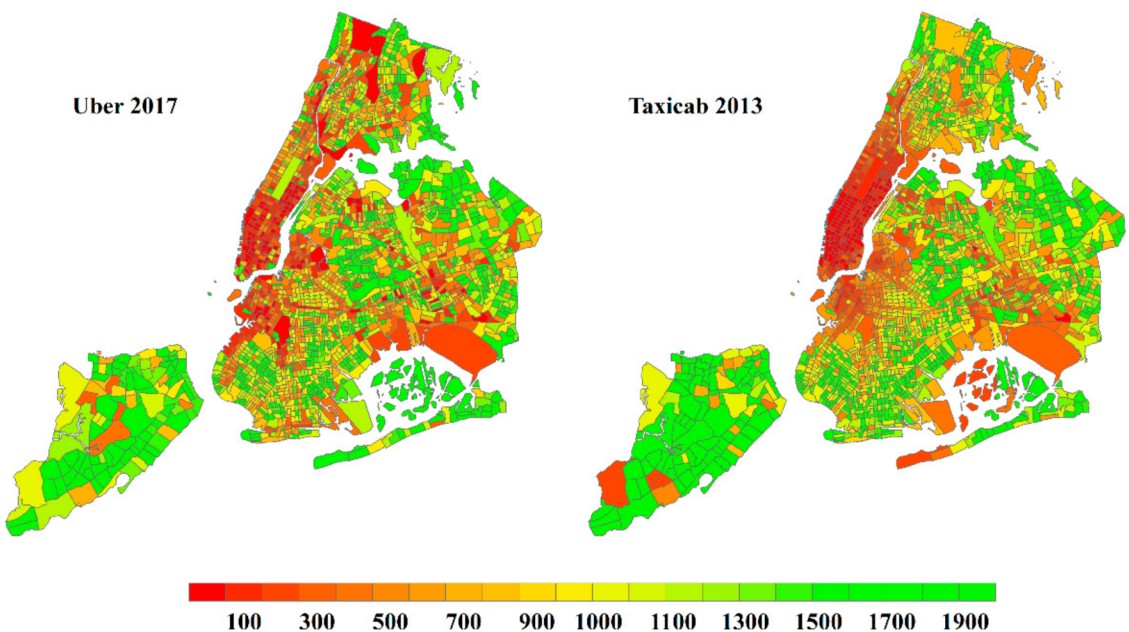

**Figure 7.** Importance of spatial units on empty flow network (From red to green: higher importance/rank to lower importance/rank).

#### 5.1.2. Spatial Scope of Search Activities

In this section, we measure gyration radius for each driver. Traditional and app-based drivers show very different distributions on gyration radius in Figure 8. Almost all traditional taxicab drivers search for their next passengers in a radius of 2 to 3 miles. However, Uber drivers have a larger searching region with a radius of 4 to 6 miles. Two factors may lead to the larger searching region: (1) a considerable number of app-based taxi drivers are part-time drivers and serve primarily around both their workplace and home with a certain distance; and (2) the smartphone application provides demand information in a larger area, which may encourage drivers to search passengers in a remote but high-demand area.

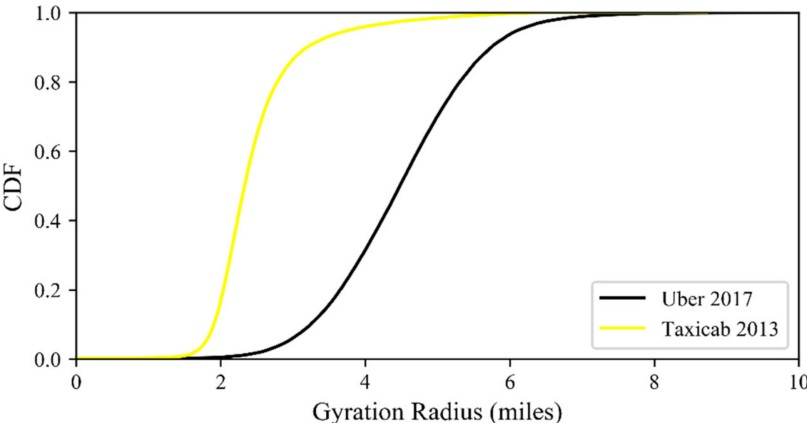

**Figure 8.** Gyration radius of drivers while searching for passengers.

### 5.2. Trip Travel Patterns

The empty traditional taxicab trips in 2013 are estimated from occupied trips. Although we know where and when those empty trips start and end, we cannot directly refer to Euclidean or Manhattan distance as empty travel distance. Since the empty vehicles may circulate around for customer searching, instead of directly driving to a destination, by most occupied trips. This section only explores the cumulative distribution of empty trip travel time and app-based taxi movement efficiency. Figure 9a shows that app-based taxi drivers are more efficient in customer searching by taking fewer empty trip travel time. Figure 9b presents the comparison between ratio of distance and ratio of time, as done in occupied trip analyses. Note that the most frequent 50,000 census tract-based OD pairs cover about 28% of empty app-based taxi trips. Most empty trips are above the diagonal line, which means app-based taxis are more efficient than Google common auto traffic. More interestingly, the ratios of real travelled distance to Google's fastest route distance are around 2, or even higher. This indicates app-based drivers may circulate around rather than directly driving to ride request locations. These circulations do not, however, increase travel time a lot and considerable empty trips can still finish customer searching efficiently. Compared with occupied app-based taxi trip movement efficiency, there is a higher percentage of trips that are below the diagonal lines, which reveals the impacts of customer searching behaviors and circulations on empty vehicle movement efficiency.

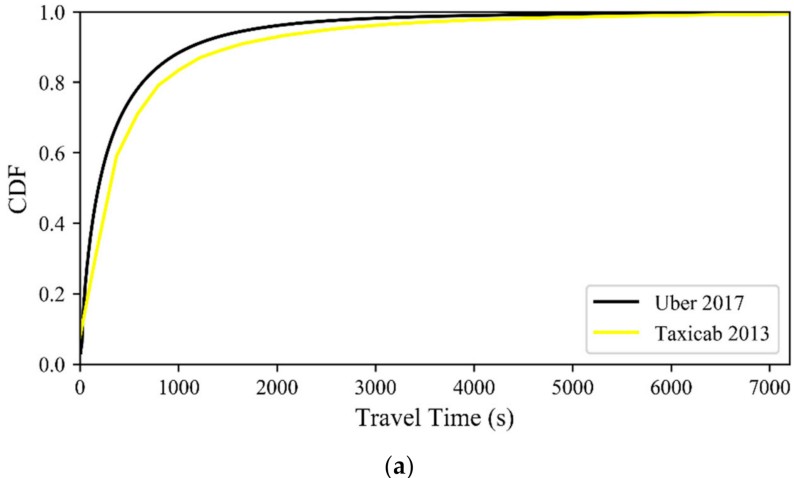

(**a**)

**Figure 9.** *Cont.*

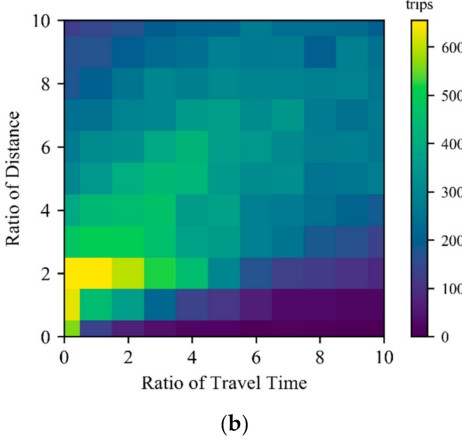

(**b**)

**Figure 9.** Cumulative distribution of travel time and passenger searching efficiency. (**a**) Cumulative distribution of empty travel time, (**b**) passenger searching efficiency.

### 5.3. Special Analyses on Uber

The detail trajectory points every two or three seconds provide us a new perspective for understanding how empty app-based taxis drive around to search passengers. Here, we randomly choose three days as representatives that are 19 April 2017 (Wednesday), 22 April 2017 (Saturday), and 28 April 2017 (Friday). In a series of trajectory points belonging to one same empty trip, cumulative distance travelled and cumulative travel time are measured at each trajectory point. Meanwhile, we also measure the Euclidean distance (called E-Distance) between the trajectory point and the empty trip origin. Moreover, instantaneous speed and ratio of cumulative distance travelled to E-Distance are estimated to detect stationary state and circulations in each empty trip. A set of sequential trajectory points with zero or very low instantaneous speeds is considered as stationary state where app-based taxi drivers may wait at somewhere instead of continuous moving. Higher ratio of cumulative distance travelled to E-Distance indicates more circulations around empty trip origins. We plot time-distance graph with instantaneous speed as third dimension (colors) and plot time-E-Distance graph with ratio of distance as third dimension (colors), shown in Figure 10.

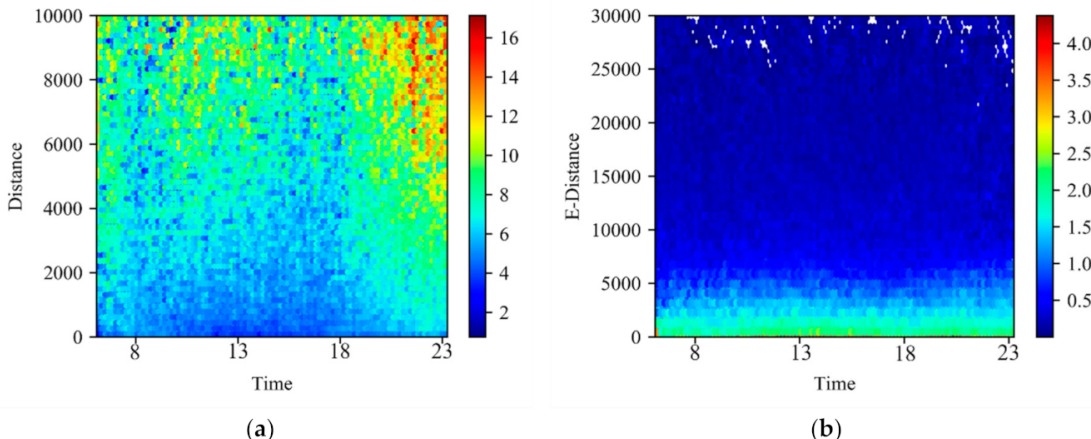

(**a**) (**b**)

**Figure 10.** *Cont.*

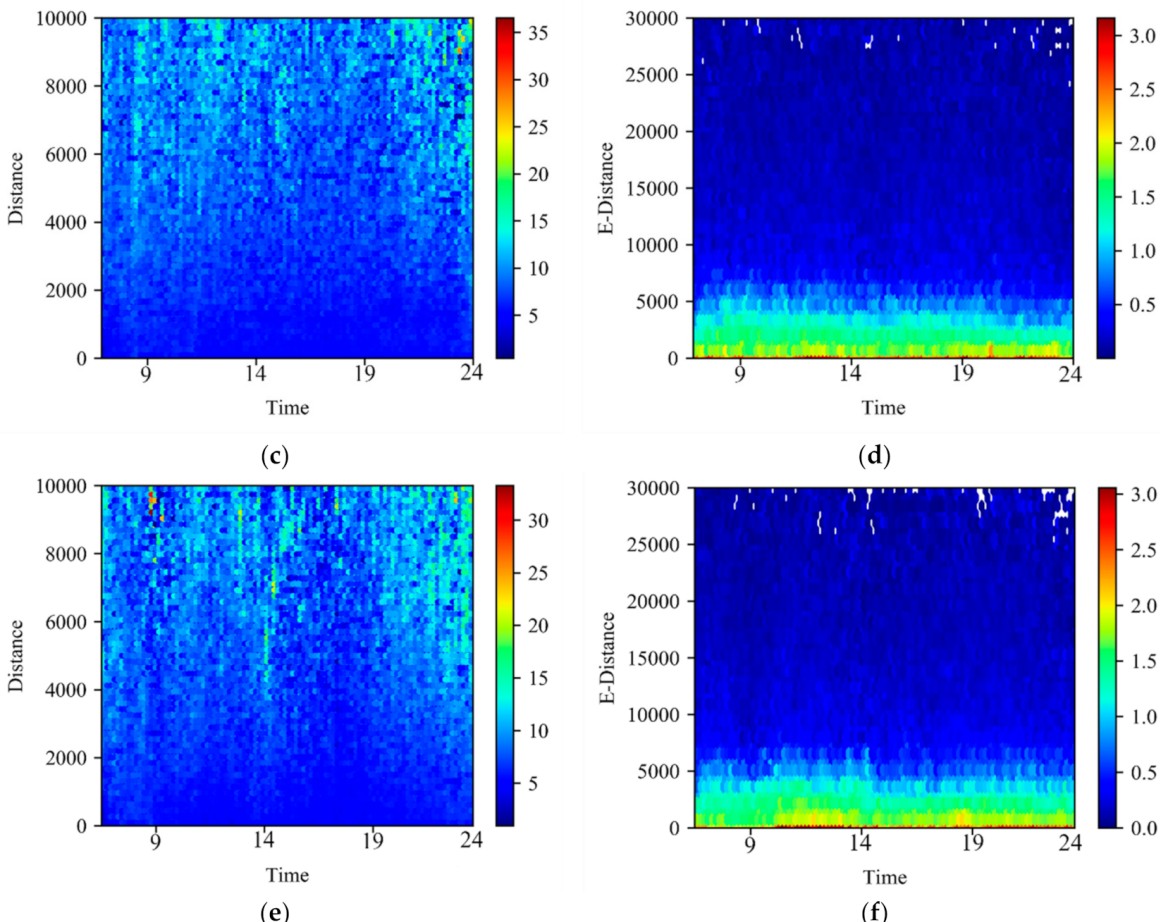

**Figure 10.** Empty Uber Trajectory. (**a**) Time-Distance on 4/19, (**b**) Time-E-Distance on 4/19, (**c**) Time-Distance on 4/22, (**d**) Time-E-Distance on 4/22, (**e**) Time-Distance on 4/28 (**f**) Time-E-Distance on 4/28.

From the three subplots at the left side, we can see that the empty trip trajectory pattern does not vary significantly across days of the week. The movement speed tends to be much lower at the beginning of empty trips than thereafter. Moreover, app-based taxi drivers may experience higher speed in the early morning or late night when compared to daylight, mainly determined by temporal variations in urban traffic. As expected, we also capture stationary state along empty trajectory (see short horizontal lines with deep blue colors in the upper of plot). In addition to the stationary state at the beginning of empty trips, Uber drivers may stop somewhere waiting for new ride requests after driving more than 1.5 miles. From the three subplots at the right side, we can also identify similar trajectory patterns across days of te week, except for fewer circulations on 19 April 2017. Generally, the circulations decrease as OD distance increases. The more distant the empty trips are, the less likely it is that the empty trips circulate around origins. The higher ratio of distance around zero or very low E-Distance indicates that considerable empty Uber trips likely circulate back to origins.

## 6. Conclusions and Discussion

This study integrates current advances in urban mobility computing techniques with our three real datasets collected from the case of Yellow taxicabs, Boro taxicabs, and UberX in NYC. Introducing various quantitative metrics allows us to explore how traditional and app-based taxi vehicles move around a city, as well as their differences. The better spatiotemporal coverage of datasets also provides multi-scale perspectives for movement explorations, including: aggregated occupied flow patterns, scaling properties of trip displacement, and trip trajectory feature extraction.

Beyond the spatial distributions of ridership that are shifting outside of downtown areas, we also identify the significant local spatial autocorrelations of intra- and inter-service rides. The network representation combined with PageRank centrality uncovers almost same spatial connections and critical hubs of both taxis other than a few remote areas of importance by app-based taxi. The gyration radius discovers the different scope of passengers and drivers' activities that passengers by app-based taxis have shorter and stable travel distances and app-based taxi drivers tend to have a greater passenger searching radius. A few scaling properties on travel time and trip distance are explored based on cumulative distribution. Although there are minor differences in occupied movements, the empty trips show different movement patterns between both taxis. Moreover, the movement efficiency is also compared with a benchmark traffic obtained from Google API. Both taxis have comparative movement efficiency as Google common auto traffic and app-based taxis show slightly higher efficiency. Finally, we leverage detailed Uber trajectory on movement feature extraction and confirm the existence of stationary/stops and circulations.

Our study is one of the first few empirical discussions on traditional and app-based taxi movement patterns, as well as their comparisons. The findings are useful for both city authority and transportation network companies. On one hand, the city authority can acknowledge the operation status of every region and develop strategy plan in the city. For example, how urban residents move around city by taxi, where are hot places for taxi services, and how does the taxi industry change after app-based service emerges. On the other hand, the transportation network companies will learn more demand pattern and trip characteristics. In particular, the patterns of unoccupied trips may help evaluate the quality of customer searching and enhance drivers' efficiency.

However, the data availability limits further in-depth explorations, although we have introduced three taxicab- and Uber-related datasets. First, we do not have app-based taxi request and ride records. The current study estimates occupied app-based taxi trips indirectly from empty trajectory data. This may introduce a few phantom occupied trips considering frequent vehicle offline or online states and complicated behaviors. Second, we do not have traditional taxi empty trip records in recent years and can only refer to an old dataset from 2013. Moreover, we do not have this year's traditional taxicab occupied trip records. However, the last one is easier to be fixed once NYCTLC releases new datasets in the next few months.

**Author Contributions:** Conceptualization, Wenbo Zhang; methodology, Wenbo Zhang and Chang Xu; software, Wenbo Zhang; validation, Wenbo Zhang; formal analysis, Wenbo Zhang; writing—original draft preparation, Wenbo Zhang and Chang Xu; writing—review and editing, Wenbo Zhang; visualization, Wenbo Zhang and Chang Xu. All authors have read and agreed to the published version of the manuscript.

**Funding:** This research was funded by National Nature and Science Foundation of China—Youth Program, grant number 52002064, and Humanities and Social Science Youth Program, Ministry of Education, China, grant number 20YJC630216.

**Data Availability Statement:** Publicly available datasets were analyzed in this study. This data can be found here: https://www1.nyc.gov/site/tlc/about/tlc-trip-record-data.page.

**Conflicts of Interest:** The authors declare no conflict of interest.

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
