# Peer review of "Exploring App-Based Taxi Movement Patterns from Large-Scale Geolocation Data"

_ijgi, doi:10.3390/ijgi10110751_

Round 1

Reviewer 1 Report

The paper explored App-based taxi movement patterns from large scale geolocation data. The paper provided a very good empirical research on traditional and app-based taxi movement patterns, as well as their comparisions. The research is interesting and important.

In the future research,  if more detailed datasest can be collected and used in the research, it will produce better research results.

Author Response

Thanks for your comments. The authors are now cooperating with authority agency in one large city of China and will get trip order and trajectory data from all ride-hailing platforms, as well as street-hailing taxicabs. The study can be further extended and explore new things even among different platforms.

Reviewer 2 Report

The idea of this research is interesting. The study can be used in a Smart City Strategy. The authors have graphic representations in a good way. The mathematical models and methods used are not new but are used well in my opinion. The results obtained are useful. The bibliography used is adequate.

However, I believe that the work can be improved. The obtained results must be related more to the reality in the field. Also in the conclusions it is necessary to specify more the utility of this research and the way in which the obtained results can generate solutions for the problems of the city.

In conclusion, the work is good and small improvements can be made.

Author Response

Thanks for your constructive advice. We agree with your point and should specify the implications of our findings. In the conclusion section of revised version, we add some descriptions on the implications and clearly present how our findings can be used in reality. You can refer to the revised version for our revisions. 

Reviewer 3 Report

First of all, thank you for submitting the article for review. The article describes analyzes based on spatio-temporal data of three taxi companies in NYC. It contains many different analysis results and their combinations. I have some comments:

  1. In the introduction, the last paragraph specifies what is included in the article. However, I do not find a clear definition of the aim of the article.
  2. The work contains a great variety of results, but there is no discussion section.
  3. The article does not answer the questions why the findings are interesting and original, and how they affect a wider understanding of the topic.
  4. Figure 2 - the last map in the first line is on a different scale. Figure would be clearer without gray borders.
  5. Figure 9 - part a and b of the drawing is divided into pages

Round 2

Reviewer 3 Report

Thanks for the changes. I have no more comments on the text.